# FAVAE: Sequence Disentanglement using Information Bottleneck Principle

## Abstract

We propose the factorized action variational autoencoder (FAVAE), a state-of-the-art generative model for learning disentangled and interpretable representations from sequential data via the information bottleneck without supervision. The purpose of disentangled representation learning is to obtain interpretable and transferable representations from data. We focused on the disentangled representation of sequential data since there is a wide range of potential applications if disentanglement representation is extended to sequential data such as video, speech, and stock price data. Sequential data is characterized by dynamic factors and static factors: dynamic factors are time dependent, and static factors are independent of time. Previous works succeed in disentangling static factors and dynamic factors by explicit modeling the priors of latent variables to distinguish between static and dynamic factors. However, this model cannot disentangle representations between dynamic factors, such as disentangling "picking" and "throwing" in robotic tasks. In this paper, we propose new model that can disentangle multiple dynamic factors. Since our method does not require modeling priors, it is capable of disentangling "between" dynamic factors. In experiments, we show that FAVAE can extract the disentangled dynamic factors.

## 1 Introduction

Representation learning is one of the most fundamental problems in machine learning. A real world data distribution can be regarded as a low-dimensional manifold in a high-dimensional space (Bengio et al., 2013). Generative models in deep learning, such as the variational autoencoder (VAE) (Kingma & Welling, 2013) and the generative adversarial network (GAN) (Goodfellow et al., 2014), are able to learn low-dimensional manifold representation (factor) as a latent variable. The factors are fundamental components such as position, color, and degree of smiling in an image of a human face (Liu et al., 2015). Disentangled representation is defined as a single factor being represented by a single latent variable (Bengio et al., 2013). Thus, if in a model of learned disentangled representation, shifting one latent variable while leaving the others fixed generates data showing that only the corresponding factor was changed. This is called *latent traversals* (a good demonstration of which was given by Higgins et al. (2016a)[1]). There are two advantages of disentangled representation. First, latent variables are interpretable. Second, the disentangled representation is generalizable and robust against adversarial attacks (Alemi et al., 2016).

We focus on the disentangled representation learning of sequential data. Sequential data is characterized by dynamic factors and static factors: dynamic factors are time dependent, and static factors are independent of time. With disentangled representation learning from sequential data, we should be able to extract dynamic factors that cannot be extracted by disentangled representation learning models for non-sequential data such as $\beta$-VAE (Higgins et al., 2016a;b) and InfoGAN (Chen et al., 2016). The concept of disentangled representation learning for sequential data is illustrated in Fig. 1. Consider that the pseudo-dataset of the movement of a submarine has a dynamic factor: the trajectory shape. The disentangled representation learning model for sequential data can extract this shape. On the other hand, since the disentangled representation learning model for non-sequential data does not consider the sequence of data, it merely extracts the x-position and y-position.

---

[1]This demonstration is available at http://tinyurl.com/jgbyzke

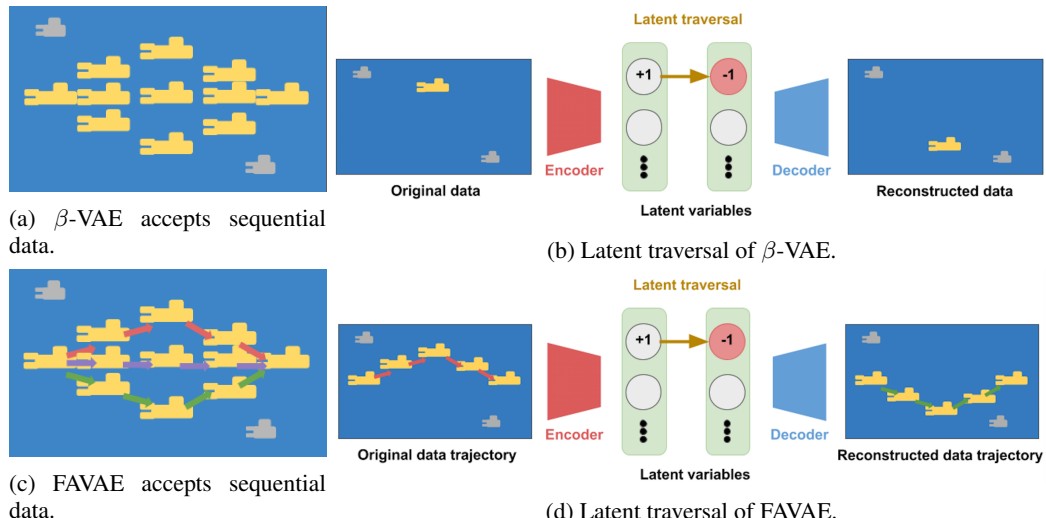

(a) $\beta$-VAE accepts sequential data.

(b) Latent traversal of $\beta$-VAE.

(c) FAVAE accepts sequential data.

(d) Latent traversal of FAVAE.

Figure 1: Illustration of how FAVAE differs from $\beta$-VAE. $\beta$-VAE does not accept data sequentially; it cannot differentiate data points from different trajectories or sequences of data points. FAVAE considers a sequence of data points, taking all data points in a trajectory as one datum. For example, for a pseudo-dataset representing the trajectory of a submarine (1a,1c), $\beta$-VAE accepts 11 different positions of the submarine as non-sequential data while FAVAE accepts three different trajectories of the submarine as sequential data. Therefore, the latent variable in $\beta$-VAE learns only the coordinates of the submarine, and the latent traversal shows the change in the submarines position. On the other hand, FAVAE learns the factor that controls the trajectory of the submarine, so the latent traversal shows the change in the submarines trajectory.

There is a wide range of potential applications if we extend disentanglement representation to sequential data such as speech, video, and stock market data. For example, disentangled representation learning for stock price data can extract the fundamental trend of a given stock price. Another application is the reduction of action space in reinforcement learning. Extracting dynamic factors would enable the generation of macro-actions (Durugkar et al., 2016), which are sets of sequential actions that represent the fundamental factors of the actions. Thus, disentangled representation learning for sequential data opens the door to new areas of research.

Very recent related work (Hsu et al., 2017; Li & Mandt, 2018) separated factors of sequential data into dynamic and static factors. The factorized hierarchical variational autoencoder (FHVAE) (Hsu et al., 2017) is based on a graphical model using latent variables with different time dependencies. By maximizing the variational lower bound of the graphical model, the FHVAE separates the different time dependent factors such as the dynamic and static factors. The VAE architecture developed by Li & Mandt (2018) is the same as the FHVAE in terms of the time dependencies of the latent variables. Since these models require different time dependencies for the latent variables, these approaches cannot be used disentangle variables with the same time dependency factor.

We address this problem by taking a different approach. First, we analyze the root cause of disentanglement from the perspective of information theory. As a result, the term causing disentanglement is derived from a more fundamental rule: reduce the mutual dependence between the input and output of an encoder while keeping the reconstruction of the data. This is called the information bottleneck (IB) principle. We naturally extend this principle to sequential data from the relationship between $x$ and $z$ to $x_{t:T}$ and $z$. This enables the separation of multiple dynamic factors as a consequence of information compression. It is difficult to learn a disentangled representation of sequential data since not only the feature space but also the time space should be compressed. We created the factorized action variational autoencoder (FAVAE) in which we implemented the concept of information capacity to stabilize learning and a ladder network to learn a disentangled representation in accordance with the level of data abstraction. Since our model is a more general model without the restriction of a graphical model design to distinguish between static and dynamic factors, it can separate depen-

dency factors occurring at the same time. Moreover, it can separate factors into dynamic and static factors.

## 2 DISENTANGLEMENT FOR NON-SEQUENTIAL DATA

$\beta$-VAE (Higgins et al., 2016a;b) is a commonly used method for learning disentangled representations based on the VAE framework (Kingma & Welling, 2013) for a generative model. The VAE can estimate the probability density from data x. The objective function of the VAE maximizes the evidence lower bound (ELBO) of $\log p\,(x)$ as

$$\log p\,(x) = \mathcal{L}_{\text{VAE}} + \underbrace{D_{\text{KL}}\,(q\,(z|x)\,||p\,(z|x))}_{\geq 0}, \tag{1}$$

where $z$ is latent variable, $D_{\text{KL}}$ is the Kullback-Leibler divergence, and $q\,(z|x)$ is an approximated distribution of $p\,(z|x)$. $D_{\text{KL}}\,(q\,(z|x)\,||p\,(z|x))$ reduces to zero as the ELBO $\mathcal{L}_{\text{VAE}}$ increases; thus, $q\,(z|x)$ learns a good approximation of $p\,(z|x)$. The ELBO is defined as

$$\mathcal{L}_{\text{VAE}} \equiv E_{q(z|x)}\,[\log p\,(x|z)] - D_{\text{KL}}\,(q\,(z|x)\,||p\,(z)), \tag{2}$$

where the first term, $E_{q(z|x)}\,[\log p\,(x|z)]$, is a reconstruction term used to reconstruct $x$, and the second term $D_{\text{KL}}\,(q\,(z|x)\,||p\,(z))$ is a regularization term used to regularize posterior $q\,(z|x)$. Encoder $q\,(z|x)$ and decoder $p\,(x|z)$ are learned in the VAE.

Next we will explain how $\beta$-VAE extracts disentangled representations from unlabeled data. $\beta$-VAE is an extension of the coefficient $\beta > 1$ of the regularization term $D_{\text{KL}}\,(q\,(z|x)\,||p\,(z))$ in the VAE. The objective function of $\beta$-VAE is

$$\mathcal{L}_{\beta-\text{VAE}} = E_{q(z|x)}\,[\log p\,(x|z)] - \beta D_{\text{KL}}\,(q\,(z|x)\,||p\,(z)), \tag{3}$$

where $\beta > 1$ and $p\,(z) = \mathcal{N}\,(0, 1)$. $\beta$-VAE promotes disentangled representation learning via the Kullback-Leibler divergence term. As $\beta$ increases, the latent variable $q\,(z|x)$ approaches the prior $p\,(z)$; therefore, each $z_i$ is pressured to learn the probability distribution of $\mathcal{N}\,(0, 1)$. However, if all latent variables $z_i$ become $\mathcal{N}\,(0, 1)$, the model cannot reconstruct $x$. As a result, as long as $z$ reconstructs $x$, $\beta$-VAE reduces the information of $z$.

## 3 PRELIMINARY: ORIGIN OF DISENTANGLEMENT

To clarify the origin of disentanglement, we will explain the regularization term. The regularization term has been decomposed into three terms (Chen et al., 2018; Kim & Mnih, 2018; Hoffman & Johnson, 2016):

$$E_{p(x)}\,[D_{\text{KL}}\,(q\,(z|x)\,||p\,(z))] = I\,(x; z) + D_{\text{KL}}\left(q\,(z)\,||\prod_j q\,(z_j)\right) + \sum_j D_{\text{KL}}\,(q\,(z_j)\,||p\,(z_j)), \tag{4}$$

where $z_j$ denotes the $j$-th dimension of the latent variable. The second term, which is called "total correlation" in information theory, quantifies the redundancy or dependency among a set of n random variables (Watanabe, 1960). $\beta$-TCVAE has been experimentally shown to reduce the total correlation causing disentanglement (Chen et al., 2018). The third term indirectly causes disentanglement by bringing $q\,(z|x)$ close to the independent standard normal distribution $p\,(z)$. The first term is mutual information between the data variable and latent variable based on the empirical data distribution. Minimizing the regularization term causes disentanglement but disturbs reconstruction via the first term in Eq. (4). The shift $C$ scheme was proposed (Burgess et al., 2018) as a means to solve this conflict:

$$-E_{q_\phi(z|x)}\,[\log p_\theta\,(x|z)] + \beta\,|D_{\text{KL}}\,(q\,(z|x)\,||p\,(z)) - C|, \tag{5}$$

where constant shift $C$, which is called "information capacity," linearly increases during training. This shift $C$ can be understood from the point of view of an information bottleneck (Tishby et al., 2000). The VAE can be derived by maximizing the ELBO, but $\beta$-VAE can no longer be interpreted as an ELBO once this scheme has been applied. The objective function of $\beta$-VAE is derived from

the information bottleneck (Alemi et al., 2016; Achille & Soatto, 2018; Tishby et al., 2000; Chechik et al., 2005).

$$\max I(z; x) \qquad \text{s.t. } |I(\hat{x}; z) - C| = 0, \qquad (6)$$

where $C$ is the information capacity and $\hat{x}$ is the empirical distribution. Solving this equation by using Lagrange multipliers drives the objective function of $\beta$-VAE Eq. (5) with $\beta$ as the Lagrange multiplier (details in Appendix B of (Alemi et al., 2016)). In Eq. (5), information capacity $C$ prevents $I(\hat{x}, z)$ from becoming zero. In the information bottleneck literature, $y$ typically stands for a classification task; however, the formulation can be related to the autoencoding objective (Alemi et al., 2016). Therefore, the objective function of $\beta$-VAE can be understood using the information bottleneck principle.

## 4 PROPOSED MODEL: DISENTANGLEMENT FOR SEQUENTIAL DATA

Our proposed FAVAE model learns disentangled and interpretable representations from sequential data without supervision. We consider sequential data $x_{1:T} \equiv \{x_1, x_2, \cdots, x_T\}$ generated from a latent variable model,

$$p(x_{1:T}) = \int p(x_{1:T}|z) p(z) dz. \qquad (7)$$

For sequential data, we replace $x$ with $(x_{1:T})$ in Eq. 5. The objective function of the FAVAE model is

$$-E_{q_\phi\left(z|(x_{1:T})_i\right)}\left[\log p_\theta\left(x_{1:T}|z\right)\right] + \beta \left|D_{\mathrm{KL}}\left(q\left(z|(x_{1:T})_i\right)||p(z)\right) - C\right|, \qquad (8)$$

where $p(z) = \mathcal{N}(0, 1)$. The variational recurrent neural network (Chung et al., 2015) and stochastic recurrent neural network (Fraccaro et al., 2016) extend the VAE model to a recurrent framework. The priors of both networks are dependent on time. The time dependent prior experimentally improves the ELBO. In contrast, the prior of our model is independent of time like those of the stochastic recurrent network (Bayer & Osendorfer, 2014) and the Deep Recurrent Attentive Writer (DRAW) neural network architecture (Gregor et al., 2015); this is because FAVAE is disentangled representation learning rather than density estimation. For better understanding, consider FAVAE from the perspective of IB. As with $\beta$-VAE, FAVAE can be understood from the information bottleneck principle.

$$\max I(z; x_{1:T}) \qquad \text{s.t. } |I(\hat{x}_{1:T}; z) - C| = 0, \qquad (9)$$

where $\hat{x}_{1:T}$ follows an empirical distribution. These principles make the representation of $z$ compact while reconstruction of the sequential data is represented by $x_{1:T}$ (see Appendix A).

### 4.1 LADDER NETWORK

An important extension to FAVAE is a hierarchical representation scheme inspired by the VLAE (Zhao et al., 2017). Encoder $q(z|x_{1:T})$ within a ladder network is defined as

$$h_l = f_l(h_{l-1}), \qquad (10)$$
$$z_l \sim \mathcal{N}(\mu_l(h_l), \sigma_l(h_l)), \qquad (11)$$

where $l$ is a layer index, $h_0 \equiv x_{1:T}$, and $f$ is a time convolution network, which is explained in the next section. Decoder $p(x_{1:T}|z)$ within the ladder network is defined as

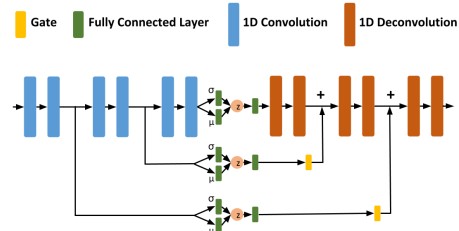

Figure 2: FAVAE architecture.

$$\tilde{z}_L = g_l(z_L) \qquad (12)$$
$$\tilde{z}_l = g_l(\tilde{z}_{l+1} + \mathrm{gate}(z_l)), \qquad (13)$$
$$x_{1:T} \sim r(x_{1:T}, \tilde{z}_0) \qquad (14)$$

where $g_l$ is the time deconvolution network with $l = 1, \cdots, L - 1$, and $r$ is a distribution family parameterized by $g_0(\tilde{z}_0)$. The gate computes the Hadamard product of its learnable parameter and

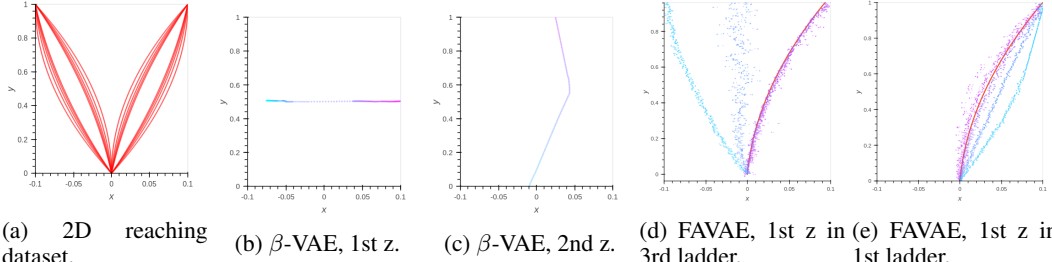

(a) 2D reaching dataset.    (b) $\beta$-VAE, 1st z.    (c) $\beta$-VAE, 2nd z.    (d) FAVAE, 1st z in 3rd ladder.    (e) FAVAE, 1st z in 1st ladder.

Figure 3: Visualization of latent traversal of $\beta$-VAE and FAVAE. On one sampled trajectory (red), each latent variable is traversed and purple and/or blue points are generated. The color corresponds to the value of the traversed latent variable. 3a represents all data trajectories of 2D reaching.

input tensor. We set $r$ as a fixed-variance factored Gaussian distribution with the mean given by $\mu_{t:T} = g_0(\tilde{z}_0)$. Fig. (2) shows the architecture of the proposed model. The difference between each ladder network in the model is the number of convolution networks through which data passes. The abstract expressions should differ between ladders since the time convolution layer abstracts sequential data. Without the ladder network, the proposed method can disentangle only the representations at the same level of abstraction; with the ladder network, it can disentangle representations at different levels of abstraction.

### 4.2 How to encode and decode

There are several mainstream neural network models designed for sequential data, such as the long short-term memory model (Hochreiter & Schmidhuber, 1997), the gated recurrent unit model (Chung et al., 2014), and the quasi-recurrent neural network QRNN (Bradbury et al., 2016). However, the VLAE has a hierarchical structure created by abstracting a convolutional neural network, so it is simple to add the time convolution of the QRNN to our model. The input data are $x_{t,i}$, where $t$ is the time index and $i$ is the dimension of the feature vector index. The time convolution considers the dimensions of feature vector $j$ as a convolution channel and performs convolution in the time direction:

$$z_{tj} = \sum_{p,i} x_{t-p,i} h_{pij} + b_j, \tag{15}$$

where $j$ is the channel index. The proposed FAVAE model has a network similar to the VAE one regarding time convolution and a loss function similar to the $\beta$-VAE one (Eq. (8)). We used the batch normalization (Hinton et al., 2012) and ReLU as activation functions though other variations are possible. For example, 1d convolutional neural networks use a filter size of 3 and a stride of 2 and do not use a pooling layer.

### 5 Measuring Disentanglement

While latent traversals are useful for checking the success or failure of disentanglement, quantification of the disentanglement is required for reliably evaluating the model. Various disentanglement quantification methods have been reported (Eastwood & Williams, 2018; Chen et al., 2018; Kim & Mnih, 2018; Higgins et al., 2016b;a), but there is no standard method. We use the mutual information gap (MIG) (Chen et al., 2018) as the metric for disentanglement. The basic idea of MIG is measuring the mutual information between latent variables $z_j$ and a ground truth factor $v_k$. Higher mutual information means that $z_j$ contains more information regarding $v_k$.

$\text{MIG} \equiv \frac{1}{K} \sum_{k=1}^{K} \frac{1}{H(v_k)} \left( I\left(z_{j^{(k)}}; v_k\right) - \max_{j \neq j^{(k)}} I\left(z_j; v_k\right) \right)$, where $j^{(k)} \equiv \arg\max_j I\left(z_j; v_k\right)$, and $H(v_k)$ is entropy for normalization. In our evaluation we experimentally measure disentanglement with MIG.

Table 1: Disentanglement scores (MIG, and Reconstruction loss) with standard deviations by 10 seeds for the different model. Best results are shown in bold.

| Model | 2D Reaching | | 2D wavy Reaching | |
|---|---|---|---|---|
| | MIG | Reconstruction | MIG | Reconstruction |
| Time Convolution AE | 0.21(27) | 0.18(13) | 0.04(3) | **0.06(1)** |
| FAVAE (without ladder and $C$) | 0.34(21) | 0.62(115) | 0.22(14) | 23.6(279) |
| FAVAE (with ladder and without $C$) | **0.41(13)** | **0.03(5)** | 0.26(7) | 0.08(2) |
| FAVAE (without ladder and with $C$) | 0.34(21) | 0.62(115) | 0.11(10) | 1.47(73) |
| FAVAE (with ladder and $C$) | **0.41(13)** | **0.03(5)** | **0.34(28)** | 1.81(170) |

## 6 RELATED WORK

Several recently reported models (Hsu et al., 2017; Li & Mandt, 2018) graphically disentangle static and dynamic factors in sequential data such as speech data and video data (Garofolo et al., 1993; Pearce & Picone, 2002). In contrast, our model performs disentanglement by using a loss function (see Eq. 8). The advantage of the graphical models is that they can control the interpretable factors by controlling the priors time dependency. Since dynamic factors have the same time dependency, these models cannot disentangle dynamic factors. A loss function model can disentangle sets of dynamic factors as well as disentangle static and dynamic factors.

## 7 EVALUATION

We evaluated our model experimentally using three sequential datasets: 2D Reaching, 2D Wavy Reaching, and Gripper. We used a batch size of 128 and the Adam (Kingma & Ba, 2014) optimizer with a learning rate of $10^{-3}$.

### 7.1 2D REACHING

To determine the differences between FAVAE and $\beta$-VAE, we used a bi-dimensional space reaching dataset. Starting from point $(0, 0)$, the point travels to goal position $(-0.1, +1)$ or $(+0.1, +1)$. There are ten possible trajectories to each goal; five are curved inward, and the other five are curved outward. The degree of curvature for all five trajectories is different. The number of factor combinations was thus 20 (2x2x5). The trajectory length was 1000, so the size of one trajectory was [1000x2].

We compared the performances of $\beta$-VAE and FAVAE trained on the 2D Reaching dataset. The results of latent traversal are transforming one dimension of latent variable z into another value and reconstructing something from the traversed latent variables. $\beta$-VAE, which is only able to learn from every point of a trajectory separately, encodes data points into latent variables that are parallel to the x and y axes (3b, 3c). In contrast, FAVAE learns through one entire trajectory and can encode disentangled representation effectively so that feasible trajectories are generated from traversed latent variables (3d, 3e).

### 7.2 2D WAVY REACHING

To confirm the effect of disentanglement through information bottleneck, we evaluated the validity of our model under more complex factors by adding more factors to the 2D Reaching dataset. Five factors in total generated data compared to the three factors that generate data in 2D Reaching. This modified dataset differed in that four out of the five factors affect only part of the trajectory: two of them affect the first half, and the other two affect the second half. This means that the model should be able to focus on a certain part of the whole trajectory and be able to extract factors related to that part. A detailed explanation of these factors is given in Appendix B.

We compared various models on the basis of MIG to demonstrate the validity of our proposed model in comparison of a time convolution AE in which a loss function is used only for the autoencoder ($\beta = 0$), FAVAE without the ladder network and information capacity $C$, and FAVAE with the ladder network and information capacity $C$. As shown in Table 1, FAVAE with the ladder network and C had the highest MIG scores for 2D Reaching and 2D Wavy Reaching. This indicates that this model learned a disentangled representation best. Note that for 2D Reaching, the best value for $C$ was small, meaning that there was little effect from adding $C$ (since the dataset was simple, this task can be solved even if the amount of information of z is small).

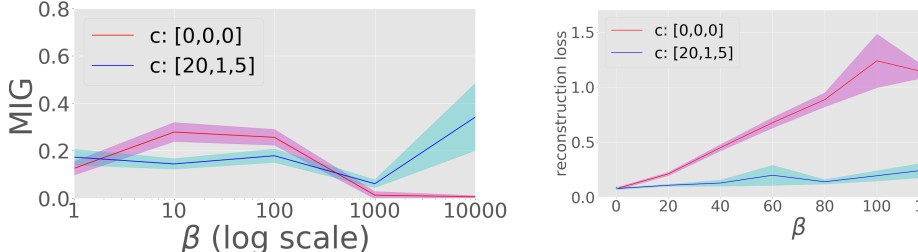

Figure 4: MIG score and reconstruction loss for different values of $\beta$. Blue line represents results with information capacity $C$ greater than zero; red line represents results with $C$ set to e zero. Note that x axis is plotted in log scale.

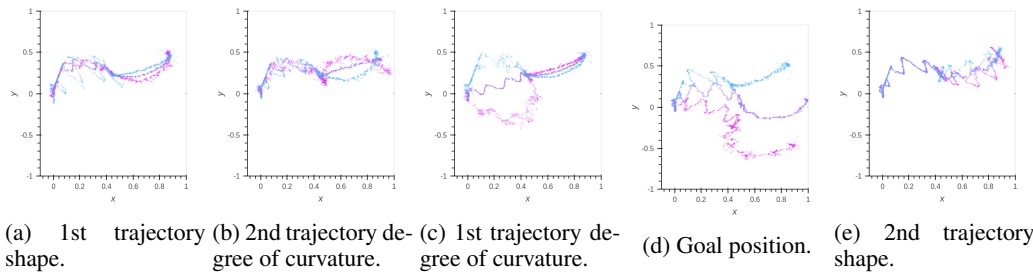

(a) 1st trajectory shape.

(b) 2nd trajectory degree of curvature.

(c) 1st trajectory degree of curvature.

(d) Goal position.

(e) 2nd trajectory shape.

Figure 5: Visualization of latent traversal for 2D Wavy Reaching. Each plot was decoded by traversing one latent variable; different colors represent trajectories generated from different values of same latent variable, z.

Fig. 4. We evaluated reconstruction loss and the MIG score for various values of $\beta$ using three ladder networks (Fig. 6) with a different value of $C$ for each ladder: $C = [\text{LowerLadder}, \text{MiddleLadder}, \text{HigherLadder}]$. One setting used was $C = [0, 0, 0]$, meaning that $C$ was not used; another setting was $C = [20, 1, 5]$, meaning that $C$ was adjusted on the basis of values of KL-Divergence for $\beta = 1$ and $C = [0, 0, 0]$. When $C$ was not used, the model could not reconstruct data when $\beta$ was high; thus, disentangled representation was not learned well when $\beta$ was high. When $C$ was used, the MIG score increased with $\beta$ while reconstruction loss was suppressed.

The latent traversal results for 2D Wavy Reaching are plotted in Fig. 5. Even though not all learned representations are perfectly disentangled, the visualization shows that all five generation factors were learned from five latent variables; the other latent variables did not learn any meaningful factors, indicating that the factors could be expressed as a combination of five "active" latent variables. We tested our model for $\beta = 300$.

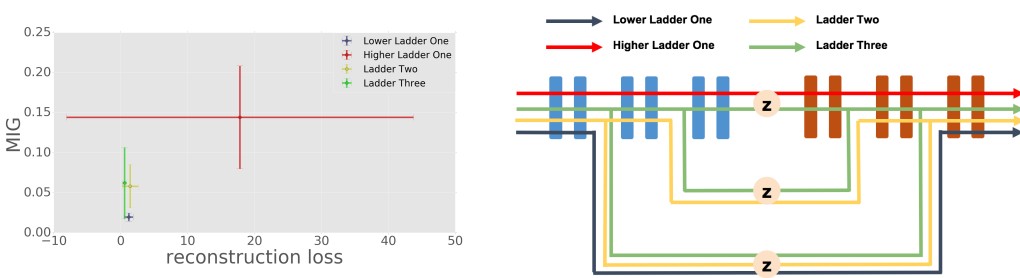

Figure 6: Evaluation of ladders (left); structure used for each experiment (right).

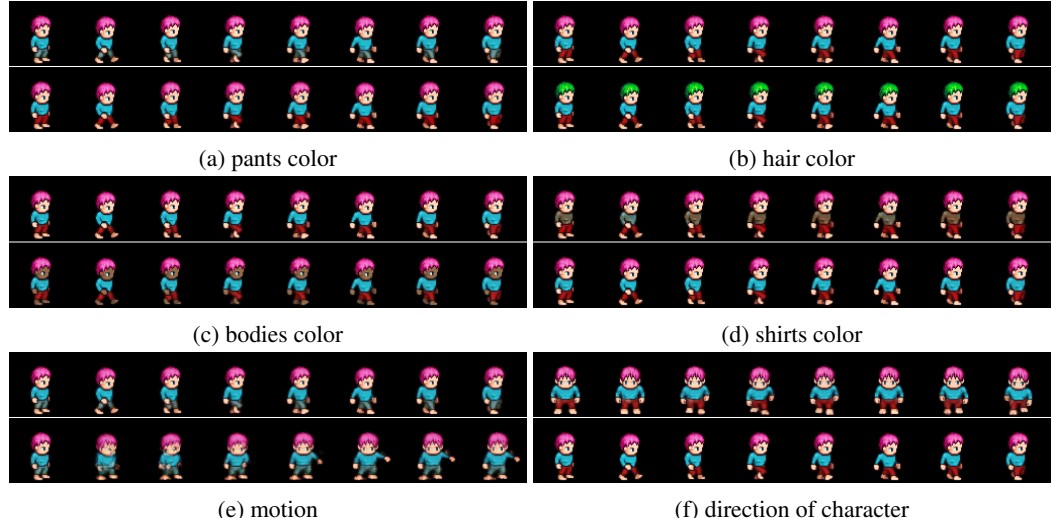

Figure 7: Visualization of latent traversal of FAVAE. The horizontal axis represents sequence and the vertical axis represents the difference of $z$ values.

The use of a ladder network in our model improved disentangled representation learning and minimized the reconstruction loss. The graph in Fig. 6 shows the MIG scores for networks with different numbers of ladders. The error bars represent the standard deviation for ten repetitive trials. Using all three ladders resulted in the minimum reconstruction loss with the highest MIG score (green curve) except for "Higher Ladder One". "Higher Ladder One" has a large reconstruction error.

### 7.3 SPRITES DATASET.

To evaluate the effectiveness of the video dataset, we trained our model with the Sprites dataset which is used in (Li & Mandt, 2018). This dataset has sequential length $= 8$ RGB video data with $3 \times 64 \times 64$. This data set consists of static factor and dynamic factor. We note that the motion is not created with the combination of dynamic factors, and each motion exists individually (detail is E.2). Table 4 show the factors used in our experiment. We executed disentangled representation learning by using the FAVAE model with $\beta = 20$, $C = [0.3, 0.17, 0.06]$ and network architecture used for this training is explained in Section F.1. Fig. 7 shows the results of latent traversal, and we chose two $z$ values to change from $z = 3$ to $3$. Since this dataset is composed of discrete factors, we show two $z$ values at a time. The latent variables in the 1st ladder extract expressions of motion (4th $z$ in 1st ladder), pants color (5th $z$ in 1st ladder), direction of character (6th $z$ in 1st ladder) and the shirts color (7th $z$ in 1st ladder). The latent variables in the 2nd ladder extract expressions of the hair color (1st $z$ in 2nd ladder) and the body color (2nd $z$ in 2nd ladder). FHVAE can extract the disentangled representations between static factors and dynamic factor in high dimension dataset.

## 8 SUMMARY AND FUTURE WORK

Our factorized action variational autoencoder (FAVAVE) generative model learns disentangled and interpretable representations via the information bottleneck from sequential data. Evaluation using three sequential datasets demonstrated that it can learn disentangled representations. Future work includes extending the time convolution part to a sequence-to-sequence model (Sutskever et al., 2014) and applying the model to actions of reinforcement learning to reduce the pattern of actions.

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

## A    Information Bottleneck Principle for Sequential Data.

Here the aim is to show the relationship between FAVAE and the information bottleneck for sequential data. Consider the information bottleneck object:

$$\max I\left(z; x_{1:T}\right) \qquad \text{s.t. } \left|I\left(\hat{x}_{1:T}; z\right) - C\right| = 0 \tag{16}$$

is expanded from Alemi et al. (2016) to sequential data. We need to distinguish between $\hat{x}_{1:T}$ and $x_{1:T}$, where $x_{1:T}$ is the true distribution and $\hat{x}_{1:T}$ is the empirical distribution created by sampling from the true distribution. We maximize the mutual information of true data $x_{1:T}$ and $z$ while constraining the information contained in the empirical data distribution. We do this by using Lagrange multiplier:

$$I\left(z; x_{1:T}\right) + \beta \left|I\left(\hat{x}_{1:T}; z\right) - C\right|, \tag{17}$$

where $\beta$ is a constant. For the first term,

$$
\begin{aligned}
I\left(z; x_{1:T}\right) &= \iint p\left(z; x_{1:T}\right) \log \frac{p\left(x_{1:T}|z\right)}{p\left(x_{1:T}\right)} dx_{1:T} dz \\
&= \iint p\left(z; x_{1:T}\right) \log p\left(x_{1:T}|z\right) dx_{1:T} dz - \iint p\left(x_{1:T}; z\right) \log p\left(x_{1:T}\right) dx_{1:T} dz \\
&= \iint p\left(x_{1:T}\right) p\left(z|x_{1:T}\right) \log p\left(x_{1:T}|z\right) dx_{1:T} dz + H\left(x_{1:T}\right) \\
&\sim \frac{1}{N} \sum_i \left[p\left(z| \left(x_{1:T}\right)_i\right) \log p\left(\left(x_{1:T}\right)_i |z\right)\right] + H\left(x_{1:T}\right),
\end{aligned}
\tag{18}
$$

where $H\left(x_{1:T}\right)$ is entropy, which can be neglected in optimization. The last line is Monte Carlo approximation. For the second term,

$$
\begin{aligned}
I\left(\hat{x}_{1:T}; z\right) &= \int p\left(z|\hat{x}_{1:T}\right) p\left(\hat{x}_{1:T}\right) \log \frac{p\left(z|\hat{x}_{1:T}\right)}{p\left(z\right)} dz d\hat{x}_{1:T} \\
&\sim \frac{1}{N} \sum_j p\left(z| \left(x_{1:T}\right)_j\right) \log \frac{p\left(z| \left(x_{1:T}\right)_j\right)}{p\left(z\right)} dz \\
&= \frac{1}{N} \sum_j D_{\mathrm{KL}}\left(p\left(z| \left(x_{1:T}\right)_j\right) ||p\left(z\right)\right).
\end{aligned}
\tag{19}
$$

As a result,

$$
\begin{aligned}
I\left(z; x_{1:T}\right) + \beta \left|I\left(\tilde{x}_{1:T}; z\right) - C\right| \leq{} & \frac{1}{N} \sum_i \left[p\left(z| \left(x_{1:T}\right)_i\right) \log p\left(\left(x_{1:T}\right)_i |z\right)\right] \\
& + \frac{1}{N} \sum_j D_{\mathrm{KL}}\left(p\left(z| \left(x_{1:T}\right)_j\right) ||p\left(z\right)\right).
\end{aligned}
\tag{20}
$$

For convenience of calculation, we use $x_i$ sampled from mini-batch data for both the reconstruction term and the regularization term. This is only an approximation. If the information bottleneck principle is followed completely, it is better to use different batch data for the reconstruction and regularization terms.

## B    Extracted factors of each ladder in FAVAE

We expect the ladder network can disentangle representations at different levels of abstraction. In this section, We check the factor extracted in each ladder by using 2D Reaching and 2D Wavy.

Table 2: Counting the index of latent variable in each factor by 10 seeds. The detail of factor is in Table 4.

|  |  | 1st ladder | 2nd ladder | 3rd ladder |
|---|---|---|---|---|
| | factor 1 | 1 | 1 | 8 |
| 2D Reaching | factor 2 | 10 | 0 | 0 |
| | factor 3 | 10 | 0 | 0 |
| | factor 1 | 3 | 0 | 7 |
| | factor 2 | 8 | 0 | 2 |
| 2D wavy Reaching | factor 3 | 8 | 0 | 2 |
| | factor 4 | 9 | 1 | 0 |
| | factor 5 | 9 | 0 | 1 |

Table 3: Disentanglement scores (MIG, and Reconstruction loss) with standard deviations by 10 seeds for the different model in sequence length $= 100$. Best results are shown in bold.

| Model | 2D Reaching | | 2D wavy Reaching | |
|---|---|---|---|---|
| | MIG | Reconstruction | MIG | Reconstruction |
| FHVAE | **0.62(17)** | **0.0003(5)** | 0.19(6) | 0.051(36) |
| Time Convolution AE | 0.06(4) | 0.014(7) | 0.26(6) | **0.019(2)** |
| FAVAE (without ladder and $C$) | 0.52(6) | 0.007(1) | 0.28(2) | 0.058(3) |
| FAVAE (without ladder with $C$) | 0.47(4) | 0.005(0) | 0.26(1) | 0.069(4) |
| FAVEA (with ladder without $C$) | 0.34(11) | 0.005(1) | 0.31(8) | 0.022(2) |
| FAVAE (with ladder and $C$) | 0.35(7) | 0.007(1) | **0.40(7)** | 0.055(4) |

Table 2 shows the counting index of latent variable with the highest mutual information in each ladder network. In Table 2, the rows represent factor and the columns represent the index of the ladder networks. The factor 1 (goal left / goal right) in 2D Reaching and the factor 1 (goal position) in 2D wavy Reaching were extracted to the most frequently in the latent variable in 3rd ladder. Since the latent variables have 8 dimensions for the 1st ladder, 4 dimensions for the 2nd ladder and 2 dimensions for the 3rd ladder, the 3rd ladder should be the least frequent when factors are randomly entered for each z. Especially long-term and short-term factors are clear in the 2D wavy Reaching dataset. In 2D Wavy Reaching dataset, there is distinct difference between factors of long and short time dependency. The "goal position" is the factor which affect the entire trajectory, and other factors affect half length of the trajectory (Fig. 9).In our experiment the goal of the trajectory which affect the entire trajectory tended to be expressed in the 3rd ladder. In both datasets, only factor 1 represents goal positions while others represent shape of the trajectories. Since factor 1 has different abstraction level from others, factor 1 and others result in different ladders such as ladder 3 and others.

## C    COMPARING WITH FHVAE

FHVAE model is the recently proposed disentangled representation learning model. We note that FHVAE model uses label information to disentangle time series data, which is different setup with our FAVAE model. Table 3 shows a comparison of MIG and reconstruction using FHVAE as the baseline. It was not possible to disentangle in 2D Reaching and 2D wavy using FHVAE, because LSTM used at the FHVAE can not learn data with very long sequences (sequence length of 1000). For fair comparison with the FHVAE, we experimented with 2D Reaching (sequence length 100), 2D wavy Reaching (sequence length 100) at Table 3. In 2D Reaching, FHVAE model has the best score, while in 2D wavy Reaching FAVAE model with ladders and $C$ has the best score. Our model is slightly worse than FHVAE in 2D Reaching Dataset. The parameters searched are $\alpha = 1$ to 30 (the best $\alpha = 20 and 1$), $\beta = 0$ to 60 (the best $\beta = 0.4$ and 4 in 2D Reaching and 2D wavy Reaching) and the best $C = [c = 0.04, 0.02, 0.02]$ and $[0.03, 0, 0, 06]$ in 2D Reaching and 2D wavy

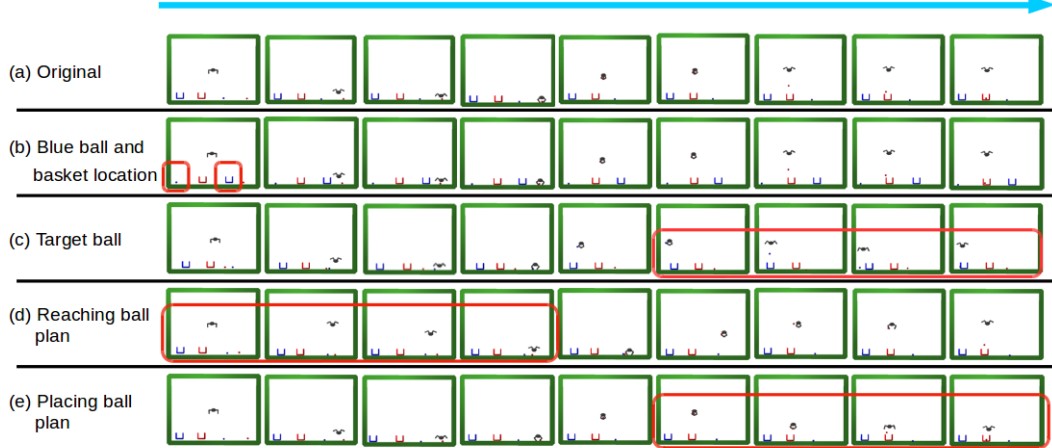

Figure 8: Visualization of learned disentangled representations for Gripper dataset. Traversed latent variable shows changing position of blue objects:(b) (9th $z$ in 1st ladder), changing target ball from red one to blue one and moving accordingly:(c) (1st $z$ in 2nd ladder), reaching ball using different plan:(d) (2nd $z$ in 2nd ladder), changing plan to place ball from drop ball to placing it gently:(e) (4th $z$ in 2nd ladder).

Reaching. The best $C$ was decided by the value of KL divergence loss when we experimented to allow reconstruction with $C = 0, 0, 0$.

## D   GRIPPER

To evaluate the potential application of our model to robotic tasks, we constructed a robot end-effector simulation environment based on Bullet Real-Time Physics Simulation[2]. The environment consisted of an end-effector, two balls (red and blue), and two baskets (red and blue) in a bi-dimensional space. The end-effector grabs one of the balls and places it into the basket with the same color as the ball. The data factors include movement "habits." For example, the end-effector could reach the targeted ball by either directly moving toward the ball obliquely or first moving above of the ball and then lowering itself until it reached the ball (perpendicular movement). The end-effector could drop the ball from far above the basket or place it gently in the basket. Each factor could affect different lengths among the data; e.g., "the plan to place the ball in the basket" factor affects different lengths per datum since the initial distance from the target ball to the corresponding basket may differ. This means that the time required to reach the ball should differ. Note that input is a value such as gripper's position, not an image. See Appendix B for a more detailed explanation.

FAVAE learned the disentangled factors of the Gripper dataset. Example visualized latent traversals are shown in Fig. 8. The traversed latent variable determined which factors were disentangled, such as the initial position of blue ball and basket (b), the targeted ball (red or blue) (c), plan to reach the ball (move obliquely or move perpendicularly) (d), and plan to place the ball in the basket (drop the ball or placing it in the basket gently) (e). These results indicate that our model can learn generative factors such as disentangled latent variables for robotic tasks, even though the length of the data affected by each factor may differ for each datum. We used FAVAE in a two-ladder network with 12 and 8 latent variables with $\beta$=1000.

---

[2]This environment is available at https://pybullet.org

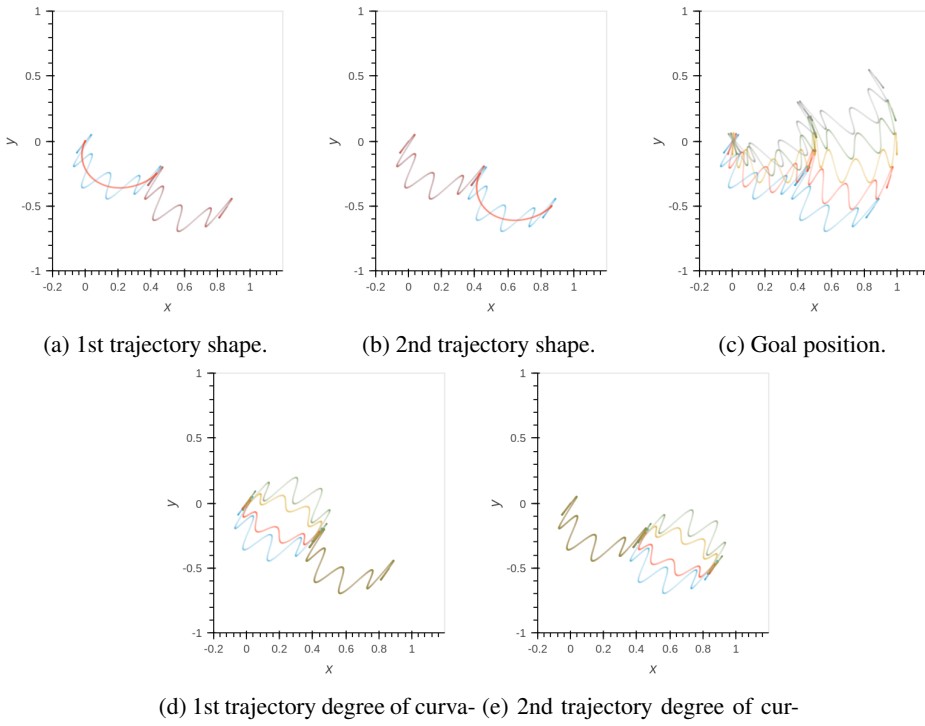

(a) 1st trajectory shape.    (b) 2nd trajectory shape.    (c) Goal position.

(d) 1st trajectory degree of curva-    (e) 2nd trajectory degree of cur-
ture.    vature.

Figure 9: Visualization of factors of 2D Wavy Reaching training data.

## E    DATASET DETAILS

### E.1    2D WAVY REACHING

Five different factors generate each trajectory: (9a) Shape of trajectory from start position to intermediate point. The first shape is created using a cycloid function while the other one adds a sine curve on the cycloid. (9b) Shape of trajectory from intermediate point to goal; shape is the same as for (9a). (9c) Location of goal; five different goal locations were used. (9d) Degree of curvature from start to intermediate point. (9e) Degree of curvature from intermediate point to goal.

### E.2    SPRITES

Sprits dataset is video data of video game "sprites". It used was used in (Li & Mandt, 2018) for confirming the extraction of disentangled representation between static factors and dynamic factors. The dataset consists of sequences with $T = 8$ frames of dimension $3 \times 64 \times 64$. We use factors and motion of Sprites is shown in Table 4 and Fig. 10.

### E.3    GRIPPER

We implemented the end-effector only rather than the entire robot arm since controlling the robot arm during the picking task is easily computable by calculating the inverse kinematics and inverse dynamics. Gripper is a 12 dimensional data set: [joint x position, joint y position, finger1 joint position(angle), finger2 joint position(angle), box1 x position, box1 y position, box2 x position, box2 y position, ball1 x position, ball1 y position, ball2 x position, ball2 y position]. Eight factors are represented in this dataset: 1) color of ball to pick up, 2) initial location of red ball, 3) initial location of blue ball, 4) initial location of blue basket, 5) initial location of red basket, 6) plan for using end effector to move to ball to pick it up [first, moving horizontally to the x-location of ball and then descending horizontally to the y-location of ball, like the movement of the doll drawing machine (perpendicular motion); second, moving straight to the location of he ball to pick it up

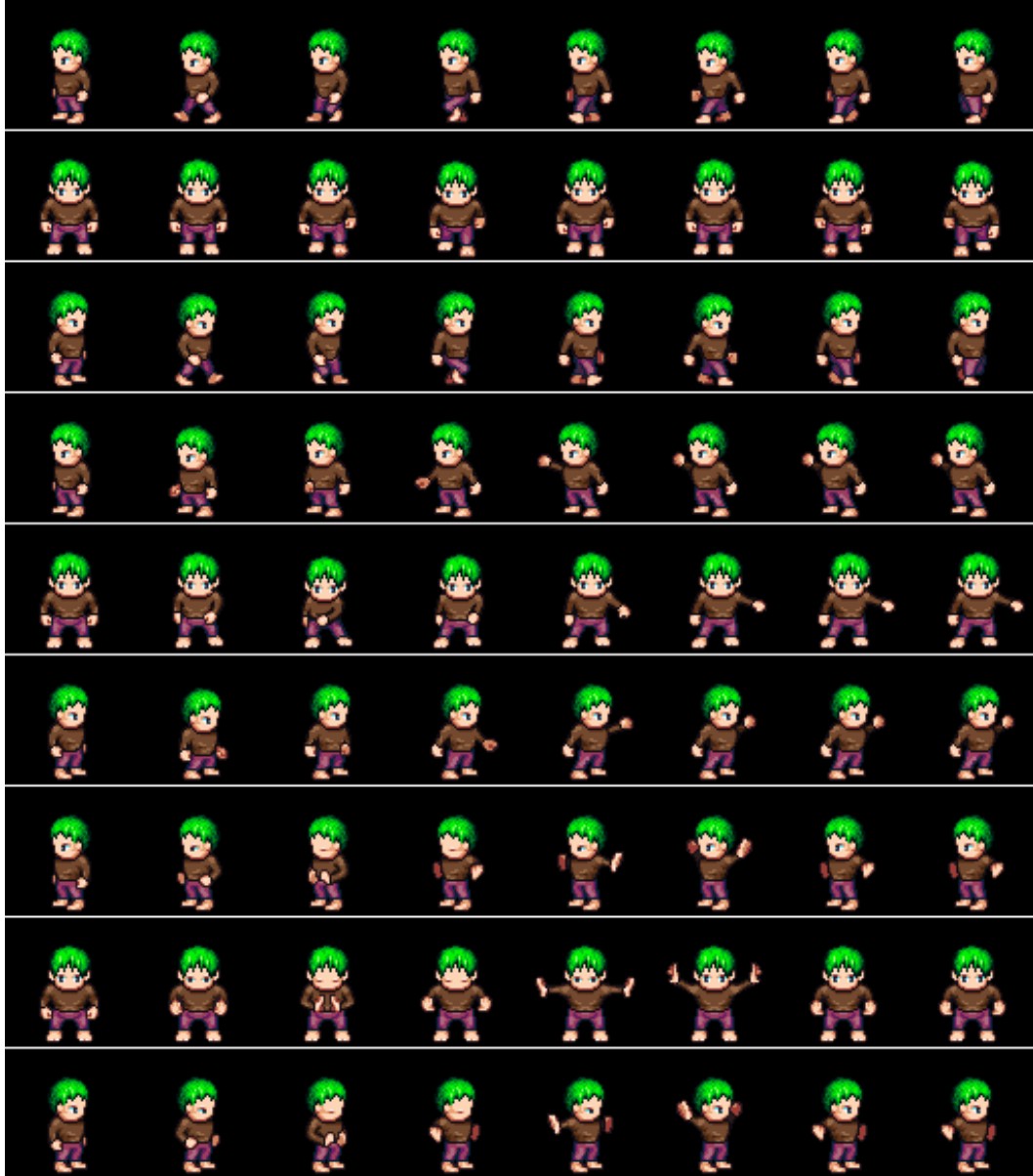

Figure 10: Motions in Sprites dataset. Row 1/2/3 represent walk left/forward/right, row 4/5/6 represent spellcast left/forward/right, and row 7/8/9 represent slash left/forward/right' respectfully.

Table 4: Factors in dataset

| Dataset | Degree of factors | Ground truth factors |
|---------|-------------------|----------------------|
| 2D Reaching | 2 | Goal positon |
| | 2 | Curved inward / outward |
| | 5 | Degree of curvature |
| 2D wavy Reaching | 4 | 1st trajectory shape |
| | 4 | 2nd trajectory shape |
| | 5 | Goal position |
| | 2 | 1st trajectory degree of curvature |
| | 2 | 2nd trajectory degree of curvature |
| Sprites Dataset | 3 | Motions (walk, spellcast or slash) |
| | 2 | Body color (light or dark2) |
| | 2 | Shirts color (brown or teal) |
| | 2 | Hair color (green or pink) |
| | 2 | Pants color (red or teal) |
| | 3 | Direction of character (left, foward, right) |
| Gripper | 2 | Targeted ball to grasp (red or blue) |
| | 4 | Red ball initial position |
| | 4 | Blue ball initial position |
| | 4 | Red basket position |
| | 4 | Blue basket position |
| | 2 | Plan to reach target ball |
| | 2 | Plan to reach corresponding basket |
| | 2 | Plan to place ball in basket |

(oblique motion)], 7) plan for using end effector to move to point above basket after picking up ball (perpendicular or oblique motion), 8) plan for placing ball in basket (by dropping ball or descending to basket and gently placing ball in basket). Among the four initial positions, the two balls and two baskets are placed randomly. The movement of the robot is hard-coded on the basis of a goal-position-based script. To reduce collision detection errors during the simulation, we used a large physical model (end effector size 1 m), which does not affect the overall validity. The length of the data-point sequence was 400.

# F ARCHITECTURE DETAILS

## F.1 FAVAE

We show the hyper parameters in Table 1, Fig. 7 and Table 3. The parameters searched are $\beta = 0$ to 10000 (the best $\beta = 1$ and $C = [0, 0, 0]$ and the best $\beta = 100000$ and $C = [20, 1, 5]$ in 2D Reaching and 2D wavy Reaching in Table 1) The best $C$ was decided by the value of KL divergence loss when we experimented to allow reconstruction with $C = 0, 0, 0$. In Table 3, we search the parameters from $\beta = 0$ to 60 (the best $\beta = 0.4$ and 4 in 2D Reaching and 2D wavy Reaching) and the best $C = [c = 0.04, 0.02, 0.02]$ and $[0.03, 0, 0, 06]$ in 2D Reaching and 2D wavy Reaching. In Fig. 7.3, we add the pre-encoder and post-decoder for encoding and decoding of images at each time step. The pre-encoding and post-decoding architecture is same as "encode_frames" and "decode_frames" in the code implementing Disentangled-Sequential-Autoencoder[3] with conv_dim $= 48$. We search the parameters from $\beta = 0$ to $\beta = 100$ and the best $\beta = 20$ with $C = [0.3, 0.17, 0.06]$.

---

[3]This code is available at https://github.com/yatindandi/Disentangled-Sequential-Autoencoder

## F.2 FHVAE

For the FHVAE experiments, we used the code from the implementation at Github[4]. We used recurrent setting with LSTM encoders and decoder with unit size 256 and batch size 80. Dimensions of latent variable z1 and z2 are 7 for each. We used Adam optimizer with learning rate 0.001. We applied goal position factors (2 classes for 2D Reaching and 5 classes for 2D wavy Reaching) as label inputs. We varied $\alpha$ from 1.0 to 30.0, and the best $\alpha$ was 20.0 for 2D Reaching and 1.0 for 2D wavy reaching.

---

[4]This code is available at https://github.com/wnhsu/FactorizedHierarchicalVAE

