# OpenReview forum: "FAVAE: SEQUENCE DISENTANGLEMENT USING IN- FORMATION BOTTLENECK PRINCIPLE"
_ICLR.cc/2019/Conference_

### Official Review · AnonReviewer3 · 2018-11-02
**Interesting, but not clear enough.**

**Rating:** 4
**Confidence:** 4

**Review:**

The paper proposes the factorized action variational autoencoder (FAVAE) as a new model for learning disentangled representations in high-dimensional sequences, such as videos. In contrast to earlier work that encouraged disentanglement through a carefully-designed dynamical prior, the authors propose a different encoder-decoder architecture combined with a modified loss function (Beta-VAE with a “shift C scheme”). As a result, the authors claim that their approach leads to useful disentangled representation learning in toy video data, and in data taken from a robotic task.

The paper appears to combine multiple ideas, which are not cleanly studied in isolation from each other. Several claims may be a bit oversold, such as potential applications for stock price data. But more importantly, the reasons why I don’t recommend accepting the paper are the following ones:

Lack of clarity:
I found that the paper lacks clarity in its presentation. Equation 7 presents a model that seems to have only a latent variable z without time dependence, but how can dynamic and static factors be separately controlled? I don't see this question addressed in the experiments. Also, what is the significance of the model architecture (ladder network) as compared to the modified loss function?  Furthermore, Fig. 7 is hard to read.

Lack of Experiments:
The currently presented experiments are all on rather simple data. I recommend extending the experiments to the Sprites data set, used in (Li and Mandt 2018), or to speech data. Also, the paper lacks comparisons to the recently proposed disentangled representation learning models (FHVAE and Disentangled Sequential Autoencoder).

While it is apparent that the model achieved some clustering, it is unclear to me if the final goal of separately controlling for static and dynamic aspects was really reached.

---

> ### Author Response · Authors · 2018-11-14
> **Reply to Reviewer 3**
>
>
> 1. > Equation 7 presents a model that seems to have only a latent variable z without time dependence, but how can dynamic and static factors be separately controlled?
>
> Since x_1: T is encoded in z, compress both static factors and dynamic factors to z. Total correlation of z decreases according to 2nd term of eq.5. That is, z is pressured to become independent. If the dynamic factor and the static factor are independent, it is possible to separate the static factor and the dynamic factor. Since FAVAE is automatically separated by pressure, separation cannot be controlled, but there is a merit that there is no need to give label information like FHVAE.
>
> 2. We reply common to all reviewers: comment 1 and 2.

---

### Official Review · AnonReviewer2 · 2018-11-03
**Interesting and valuable model extension, but with rather early results**

**Rating:** 6
**Confidence:** 5

**Review:**

This paper proposes an extension to VAE to model sequential datasets and extract disentangled representations of their evolution.
It consists of a straight extension of CCI-VAE (Burgess et al 2018) to accept sequential data, combining it with a Ladder VAE architecture (VLAE, Zhao et al 2017).
They show early results on fitting toy domains involving 2D points moving in space (reaching, reaching in sub-sequences with complex dynamics, gripper domain).

Overall, I think this is an interesting piece of work, which proposes a good model extension and assessment of its characteristics. The model is well presented, the different components are sufficiently motivated and they perform just enough experiments to showcase the effectiveness of their method, although with some reservations.

Critics:
1.	The comparison to Beta-VAE is a straw man, and I’m not sure it’s a valid way to introduce your model. You are basically saying that treating sequential data as if it was non-sequential is bad, which is clearly not surprising? Hence any comparison with Beta-VAE that you show, Figure 1 and Figure 3, are not appropriate (the caption of Fig 1 is particularly bad in that aspect). A more correct comparison would be to directly feed x_1:T to a Beta-VAE and see what happens, maybe with a causal time-convolution as well if you want to avoid 3D filters.
2.	You are also not comparing to the FHVAE model you present in your Introduction, which would have been nice to see, given that your model is simpler and requires less supervision. Does FAVAE perform better than these?
3.	Section 3 could use a citation to Esmaeili et al 2018, which breaks out the ELBO even further and compares multiple models in a really nice way (e.g. Table A.2). Overall Section 2 and 3 feel a bit long and pedantic, you could just point people to the original papers and move some of the justification to Appendix (e.g. the IB arguments are not that required for your model. ).  The main point you want to put across is that you want to have your “z” compress a full trajectory x_1:T, under a single KL pressure (i.e. last sentence of Section 4).
4.	Figure 3 was hard to interpret at first, specifically for panels b and c. Maybe if you showed the “sampled trajectory” only once in another plot it would make it clearer.
5.	Time-convolution seems to wrongly be using the opposite indexing? With z_tk = \sum_{p=0}^{H} x_{t+p}, you have an anti-causal filter which looks at the future x_t’ for a z_t? That does not sound right? Also, you should call these “causal convolutions”, which is the more standard term.
6.	The exact format of the observations was never clearly explained. From 7.1 I understood that you input 2D positions into the models, but what about the Gripper?  As you’re aware, Beta-VAE and others usually get RGB images as inputs, hence you should make that difference rather clear. This is a much simpler setting you’re working in.
7.	Did you anneal the C as was originally proposed in Burgess et al 2018? With which schedule? This was not clear to me. The exact choices of C for the different Ladder levels lacked support as well. An overall section in the Appendix about the parameter ranges you tried, the architectures, the observation specifications, the optimisation schedule etc etc would be useful.
8.	I appreciate the introduction of the MIE metric, which seems to slightly improve over MIG in a meaningful way. However, it would be good to show situations where the two metrics disagree and why MIE is better, because in the current experiments this is unclear.
9.	Overall the Gripper experiments seem to merit a more complete assessment. Figure 7 was hard to understand, and I am not sure it shows clearly any disentangled factors. Its caption was strange too (what are the “(1, 8)”, “(2, 1)” things referring to?).
10.	I would have liked more interpretation and comments on why the Ladder is needed, and why FAVAE (without Ladder and C) does so badly in Table 1.
11.	It would be good to know if you really find that the different levels of the Ladder end up extracting different time scales, as you originally claim it can. There are no results supporting this assumption in the current version.
12.	Figure 4B uses a bad scale, which makes it hard to assess what happens between the two C conditions for Beta \approx 100, where they seem to differ the most in Fig 4A.
13.	Figure 5 could use titles/labels indicating which “generative factors” you think are being represented. Just compare them to your Figure 8 in Appendix.
14.	Figure 6 MIE scores look all within noise between models considered. How sensitive is the metric to actual differences in the disentanglement?

Overall, I think this is an interesting improvement to disentangled representations learning, but suffers a bit from early experimental results. I would still like it to be shown at ICLR though as it really fits the venue.
I'm happy to improve my score given some improvements on the points mentioned above.

References:
-	Burgess et al., 2018: https://arxiv.org/abs/1804.03599
-	Zhao et al., 2017: https://arxiv.org/abs/1702.08396
-	Esmaeili et al., 2018: https://arxiv.org/abs/1804.02086

---

> ### Author Response · Authors · 2018-11-14
> **Reply to Reviewer 2**
>
>
> 1) We reply common to all reviewers: comment 1. We show Fig. 1 and 3 to clarify that the FAVAE extract disentangled representations and beta-VAE extract disentangled representations are different. The baseline model in quantitative evaluation experiment is compared with sequential model(Time convolution AE).
> 2) reply common to all reviewers:1.
> 3) > The main point you want to put across is that you want to have your “z” compress a full trajectory x_1:T, under a single KL pressure (i.e. last sentence of Section 4).
>
> We agree with the advice. We will modify for space.
>
> 4) Since sample trajectory is confusing, we will delete it in Fig. 3.
> 5) Thank you for pointing out the mistake. We modified eq. 15. Our model does not use causal convolution, 1D convolution is used. The reason why Causal convolution is not used is because 1D conv is reasonable as it can use the information of the previous and subsequent time steps. Also we do not need to use x_t for recurrent at generation, so we can use without causal convolution.
>
> 6) We use the value such as position in Gripper dataset, not the image. We added information on Gripper input x to Appendix B.2. We should clarify that there is a difference between the image and position input, so we added to Sec. 7.3 that we don't use images as input.
> 7) Yes, The scheduling c is originally proposed in Burgess et al 2018. We linear scheduling from 0 to target c as [20,1,5] in 10000 step (all experiments same). We used the same C in the same ladder(e.g. 1st ladder has 8 z dimension, it's c=20. 2nd ladder has 4 z dimension, it's c=1, 3rd ladder has 2 z dimension, it's c=5)
> 8) We reply common to all reviewers: comment 4.
> 9) We improve the cation in Fig. 7.
> 10) C is an indicator of how much information is left when compressing data. Since 2D Reaching was small in dataset information, the optimum value of C was almost 0. So for simple data FAVAE will be not bad results with C = 0.
> 11) We reply common to all reviewers: comment 3.
> 12) Does it mean that it is easier to understand by plotting in detail about β = 100 in Fig. 4B?
> 13) Added explanation of factor of 2D wavy Reaching dataset in Fig. 5.
> 14) > Figure 6 MIE scores look all within noise between models considered. How sensitive is the metric to actual differences in the disentanglement?
> In Fig. 6, only at higher ladder 1 have large error case (there was a case like loss> 57 although loss = 0.8 in general). We show the average of loss and MIG in each case (7 out of 10 succeeded).
>
> ||loss|MIG|
> |:--:|:--:|:--:|
> |success(7)|0.85|0.17|
> |fail(3)|57.3|0.071239|

---

> > ### Comment · AnonReviewer2 · 2018-11-24
> > **Good improvements!**
> >
> > Thanks to the authors for their responses and hard work in improving their results.
> >
> > 1)	The new versions of sections 2, 3 and 4 are much improved, great work!
> > 2)	I still think Figure 1 is too much space for not a very important point to make, no-one should expect Beta-VAE in its standard form to extract temporal information. It is less problematic now however.
> > 3)	Figure 5 is now clearer, thanks. Figure 4 is nicer too, but I’m not extremely sure what point you’re trying to put across with it. Does it inform your choice of beta? It is not clear from reading the text.
> > 4)	I think your results on the Sprites dataset are really promising, and actually fit the paper much better than your previous Gripper experiment. They do seem early though, so I’d recommend continuing on them, and try to give a better overview of the effect of the latents, and of which levels of “semantic factors” are captured by the different ladder levels. I think with some work and clean up they could make really strong points.
> > 5)	Related to that, I think it is still unclear in the current version exactly what is the effect/responsibilities of each ladder layer. Fig 7 goes in the good direction, but I was confused by Table 2 in Appendix B, it does not really tell a simple story of “highest/slowest ladder controls the longer-term/constant factors, vs lowest ladder controls the details of the trajectory”. At least the text does not currently express that simply.
> >
> > Overall, I think this is a nice contribution, but it would still use some more work to get it to an appropriate level for publication.

---

> > > ### Author Response · Authors · 2018-11-26
> > > **Thank you for kindly comments.**
> > >
> > > > 3) Figure 5 is now clearer, thanks. Figure 4 is nicer too, but I’m not extremely sure what point you’re trying to put across with it. Does it inform your choice of beta? It is not clear from reading the text.
> > >
> > > In the 3rd paragraph in Sec 7.2, we state that it is better to use C because when increasing Beta without introducing C, reconstruction loss gets bigger.
> > >
> > > > 4)	I think your results on the Sprites dataset are really promising, and actually fit the paper much better than your previous Gripper experiment. They do seem early though, so I’d recommend continuing on them, and try to give a better overview of the effect of the latents, and of which levels of “semantic factors” are captured by the different ladder levels. I think with some work and clean up they could make really strong points.
> > >
> > > We added the explanation how the semantic factors were extracted into certain ladders in the Sprite dataset experiments. However, as far as we tried in our experiments, we couldn't see the tendency that certain factor concentrates in certain ladder. We think that this is because it is not clear that Sprite dataset is composed of multiple abstract levels of dynamic factors.
> > >
> > >
> > > > 5)	Related to that, I think it is still unclear in the current version exactly what is the effect/responsibilities of each ladder layer. Fig 7 goes in the good direction, but I was confused by Table 2 in Appendix B, it does not really tell a simple story of “highest/slowest ladder controls the longer-term/constant factors, vs lowest ladder controls the details of the trajectory”. At least the text does not currently express that simply.
> > >
> > > We modified the texts in Appendix B. Here we state that the long time dependency is expressed in the 3rd ladder which passes time convolution the most, and short time dependency is expressed in the 1st ladder. In 2D Wavy Reaching dataset, there is distinct difference between factors of long and short time dependency. The goal of the trajectory is the factor which affect the entire trajectory, and other factors affect half length of the trajectory (Fig. 9). In our experiment the goal of the trajectory which affect the entire trajectory tended to be expressed in the 3rd ladder.

---

### Official Review · AnonReviewer1 · 2018-11-05
**Good potential but needs more work**

**Rating:** 5
**Confidence:** 4

**Review:**

This paper presents a new approach to learning disentangled representations of sequential data. The model, FAVAE, is based on the information bottleneck framework (Alemi et al, 2016; Achille et al, 2016) and extends the recent beta-VAE (Higgins et al, 2017) and CCI-VAE (Burgess et al, 2017) work by changing the encoder/decoder to a Quasi-Recurrent Neural Network (QRNN) and adding multiple latents through a ladder VAE approach (Zhao et al, 2017). The authors demonstrate that their approach is able to learn a more disentangled representation than the limited set of baselines on three toy datasets.

The authors address a very important problem, since the ability to learn disentangled representations of sequential data can have far reaching consequences, as the authors rightfully point out in their introduction. I also like the approach that the authors are taking, which appears to be very general and does not seem to involve the need to have any domain knowledge. However, the paper could be improved by clarifying certain key parts and extending the experimental section, which is currently not quite as convincing as I would have hoped.

The summary of my concerns is the following:

I would like to see more baseline comparisons 1) to an FAVAE with a different recurrent encoder/decoder; 2) to at least one other approach to disentangled representation learning on sequences; 3) an FAVAE without the capacity increase. I would also like to see all the baselines run on a non toy dataset of video data. Finally, I would like to see an expanded discussion of what the different latent variables at the different levels of the ladder architecture are learning. I recommend that the authors remove the MIE metric and shorten Section 3 to make space for the expanded experiments section.

I do hope that the authors are able to address my concerns, because their method has a lot of potential and I am excited to see where they take it during the rebuttal period. Please see the more detailed comments below:

1) Section 4.2 should be expanded to include a more detailed description of QRNN. This is one of the key modifications of FAVAE compared to CCI-VAE or beta-VAE, and it is currently not clear how QRNN works unless one reads the original paper referenced. The current paper needs to be self contained. It would also be nice to get a better understanding why QRNN was chosen over an LSTM or a GRU. It would be useful to see the results of the baseline experiments with an LSTM compared to the equivalent QRNN-based version of FAVAE.

2) Why do the authors introduce the new MIE metric? The reported results do not show a significant problem with the MIG metric, and the need for the new MIE metric is not well motivated. If the authors insist on introducing this new metric, it is important to demonstrate cases where MIG fails and MIE performs better. Otherwise I would advise removing the new metric and using the space to expand section 4.2 instead.

Another point on the metric, Eq. 16 seems to be missing a term that goes over latents z_j. I assume there should be either a max_j or a mean_j in front of the I(z_j; v_k) term in the first part of Eq. 16?

3) The related work section should be re-worded. Currently it reads as if the other approaches do not optimise a loss function at all. It would also be good to include one of the mentioned models as a baseline, and to run both FAVAE and one of the previously introduced models for disentangled sequential representation learning on a more complex dataset of video data.

4) In section 7.1, it would be good to expand the discussion of how latent traversal plots are obtained. In particular, I do not understand how the different latent variables in the ladder architecture of FAVAE are traversed. In general, it would also be nice to expand the discussion of what the different latents at the different ladder stages learn, and how the number of ladder stages affects the nature of learnt representations.

5) In terms of the baselines, it would be good to see the full ablation study. The way I see it, FAVAE has three modifications on the standard variational approaches: 1) the use of a recurrent encoder/decoder (R); 2) the use of the ladder architecture (L); and 3) the use of the capacity constraint objective (C). Currently the authors show the results of the R--, R-C, and RLC conditions. I would also like to see the results of the RL- condition (where beta=1 and C=0).

6) In terms of the results presented in Tbl. 1, it would be nice to include the hyperparameters that the authors have swept over in the appendix, as well as a table of the architecture parameters and the best hyperparameters found for the models presented in the Experiments section. In Tbl.1 it is currently unclear how many seeds the authors used to calculate the standard deviations. The units of the standard deviations presented are also not clear. Finally, it is not clear whether the differences presented in the table are significant.

7) It would be useful to include some details of the 2D Wavy Reaching dataset in the main text, even if it is just listing the nature of the 5 factors.

8) It would be useful to expand the section that talks about the different settings of C explored (page 7, paragraph 2). Currently the point that the authors are trying to make in that paragraph and Fig. 4 is not clear. I would also recommend having beta in Fig. 4 on linear rather than log scale, as the log scale seems to be somewhat confusing.




Minor points:

-- Page 2/paragraph 2: used disentangle -> used to disentangle
-- P4/p5: FAVAE is disentangled representation -> is for(?) disentangled representation
-- P6/p1: the priors time dependency -> the priors' time dependency
-- P7/p1: scores for2D -> scores for 2D
-- P7/p4: MIE score (gray curve) -> MIE score (green(?) curve)

---

> ### Author Response · Authors · 2018-11-14
> **Reply to Reviewer 1**
>
>
> 1) We would like to compare with LSTM and GRU case, but we failed to reconsturct timeseries data with recurrent neural network, so we could not make meaningful comparison.
> The reason for using time convolution is to combine several models with ladder + time conversion + CCI-VAE or beta-VAE, so we want to simplify time series processing.
>
> 2) We reply common to all reviewers: comment 4.
> 3) We reply common to all reviewers: comment 2.
> 4) We reply common to all reviewers: comment 3.
> 5) We agree with the opinion that it is important to estimate the effects of beta, c and ladder individually. We add beta=1, C=0 case in Table 1.
> 6) We add information of standard error in Table1 (the number of seed is 10). Unfortunately, we could not discuss statistically significant differences here.
> 7) Added explanation of factor of 2D wavy Reaching dataset in Fig. 5.
> 8) Does it mean that it is easier to understand by plotting in detail about β = 100 in Fig. 4B?
> We used logarithmic scales because we wanted to claim the result that reconstruction worsens as β is increased and MIG worsens as a result, unless c is adjusted to the optimum value. If the figure on the linear scale near β = 100 is better, update Fig. 4B (Should we also update fig 4A?).

---

### Author Response · Authors · 2018-11-14
**Reply common questions to all reviewers[1]**

Thank you for a lot of constructive comments. We want to provide information as much as possible.

1. On the concern that the baseline model is weak.

We will update the Table1 by using FHVAE model.
It was not possible to disentangle in 2D Reaching and 2D wavy using Baseline. This is because lstm is used in the model of FHVAE and it can not learn a very long sequence length (sequence length 1000).

For a fair comparison with the baseline we are re-experimenting on Table 1 with 2D Reaching (sequence length 100), 2 Dwavy (sequence length 100). Note that the optimal hyperparameter of FAVAE also changed in sequence length 100, so it will be described in the Appendix.


2. About experiments on more complicated data sets as video.

We are experimenting with the recommended Sprites data set, used in (Li and Mandt 2018). We'd like to add results if we can make it in time.

3. About a role of ladder network

We will add results to check that the Ladder network is doing different abstraction of time.

4. About MIE metric.

We introduced MIE to avoid the problem that MIG becomes very low value when one factor is decomposed into two latent variables. This is because MIG measures only the difference between the top two latent variables with the highest mutual information. But in our experiment MIG did not have any problem now so wel deleted the MIE section for space.

---

> ### Author Response · Authors · 2018-11-24
> **Reply common questions to all reviewers[2]**
>
> > 1. On the concern that the baseline model is weak.
> > We will update the Table1 by using FHVAE model.
>
> We added FHVAE baseline comparison in Appendix C.
> For 2D Reacing, FHVAE shows better MIG result with large error and in 2D Wavy Reaching FAVAE showed better MIG result than FHVAE even without label supervision input.
>
> > 2. About experiments on more complicated data sets as video.
> > We are experimenting with the recommended Sprites data set, used in (Li and Mandt 2018).
>
> We added experiment with Sprites dataset as recomended. While this dataset has several motions, they are not composed of multiple explicit dynamic factors. This dataset was used for extracting disentangled representation between static factors and dynamic factors in the Sprites dataset, used in (Li and Mandt 2018). We confirmed that our model can disentangle static and dynamic factors using this dataset.
>
> > 3. About a role of ladder network
> > We will add results to check that the Ladder network is doing different abstraction of time.
>
> We confirmed how FAVAE acquires disentangled representations at different levels of abstraction and added the results in Appendex B.
> The factor like goal position tends to appear in ladder3 and factor like trajectory shape tends to appear in ladder 1 and 2.

---

### Meta-Review · Area_Chair1 · 2018-12-10
**Close, but a few foundational issues, and too many major changes to re-evaluate**

**Confidence:** 3
**Recommendation:** Reject

**Metareview:**

This paper introduces an autoencoder architecture that can handle sequences of data, and attempts to automatically disentangle multiple static and dynamic factors.

Quality:  The main idea is relatively well-motivated.  However the motivation for the particular technical choices made seems a little lacking, and the complexity of the proposed model put a lot of strain on the experiments.  A lot of important updates were made by the authors in the rebuttal period, however I feel the number of changes are a lot to ask the reviewers to re-evaluate.

Clarity:  The English of the paper isn't great, including the title (should be "Using an ..." or "Using the ...").  The intro is clear enough, but belabors a relatively simple point about how an image model can't model factors in video.  There were some concerning parts where major issues seemed to be glossed over.  E.g. "FHVAE model uses label information to disentangle time series data, which is different setup with our FAVAE model."  As far as I understand, they both are trained from unsupervised data.

Originality:  This paper does a good job of citing related work, but seems incremental in relation to the FHVAE.  But the main problem is that the proposed method makes a lot of changes from a standard time-series VAE, and the limited number of experiments means it's hard to say what the important factor in this model's performance is.

Significance:  Ultimately it's hard to say what the takeaway from this paper is.  The authors motivated and evaluated a new model, but the work wasn't done in a systematic enough way to make an strong conclusions.  What conclusion were asserted seem specious and overly general, e.g. " Since dynamic factors have the same time dependency, these models cannot disentangle dynamic factors.".  Why not?  Why can't a dynamic model learn the time-scales of each of its factors automatically?